



# TPRoGI: a comprehensive rock glacier inventory for the Tibetan Plateau using deep learning

**Zhangyu Sun[1], Yan Hu[1,2*], Adina Racoviteanu[3], Lin Liu[1,2], Stephan Harrison[4], Xiaowen Wang[5], Jiaxin Cai[5], Xin Guo[5], Yujun He[5], and Hailun Yuan[5]**

[1]Earth and Environmental Sciences Programme, The Chinese University of Hong Kong, Hong Kong, China

[2]Institute of Environment, Energy and Sustainability, The Chinese University of Hong Kong, Hong Kong, China

[3]Université Grenoble Alpes, CNRS, IRD, IGE, Saint-Martin-d'Hères, France

[4]Faculty of Environment, Science and Economy, University of Exeter, Exeter, United Kingdom

[5]Faculty of Geosciences and Environmental Engineering, Southwest Jiaotong University, Chengdu, China

**Correspondence:** Yan Hu (huyan@link.cuhk.edu.hk)

**Abstract.** Rock glaciers — periglacial landforms commonly found in high mountain systems — are of significant scientific value for inferring the presence of permafrost, understanding mountain hydrology, and assessing climate impacts on high mountain environments. However, inventories remain patchy in many alpine regions, and as a result they are poorly understood for some areas of High Mountain Asia such as the Tibetan Plateau. To address this gap, we compiled a comprehensive inventory of rock glaciers across the entire Tibetan plateau, i.e., TPRoGI [v1.0], developed using an innovative deep learning method. This inventory consists of a total of 44,273 rock glaciers, covering approximately 6,000 km$^2$, with a mean area of 0.14 km$^2$. They are predominantly situated at elevations ranging from 4,000 to 5,500 m.a.s.l., with a mean of 4,729 m.a.s.l.. They tend to occur on slopes with gradients between 10° and 25°, with a mean of 17.7°. Across the plateau, rock glaciers are widespread in the northwestern and southeastern areas, with dense concentrations in the Western Pamir and Nyainqêntanglha, while they are sparsely distributed in the inner part. Our inventory serves as a benchmark dataset, which will be further maintained and updated in the future. This dataset constitutes a significant contribution towards understanding, future monitoring and assessment of permafrost on the Tibetan Plateau in the context of climate change.

## 1 Introduction

The Tibetan Plateau, the highest and largest plateau on Earth, is experiencing more pronounced warming than the global average. Currently, the warming rate on the plateau is 0.031 °C per year, higher than the global rate of 0.014 °C per year since the 1960s (Zhang et al., 2020). Moreover, areas underlain by permafrost on the plateau have experienced an even higher warming rate of 0.05°C per year since 2004 (Zhao and Sheng, 2019). This accelerated warming trend has led to rapid degradation of permafrost, which is manifested in increasing ground temperature, decreasing permafrost area, thickening active



layer, and increasing occurrence of thermokarst lakes and thaw slumps (Zhao et al., 2020; Mu et al., 2020). A valuable indicator of permafrost comes in the form of rock glaciers, defined as "*debris landforms generated by the former or current creep of frozen ground (permafrost), detectable in the landscape with the following morphologies: front, lateral margins and optionally ridge-and-furrow surface topography*" (RGIK, 2023). These landforms are widespread across the plateau, especially in the
mountainous regions.

The understanding of rock glaciers within the scientific community has been evolving since the publication of Spencer (1900)'s initial article on "a peculiar form of talus." Over the past century, the identification of rock glaciers has been the subject of ongoing debate, and the criteria for identifying them has evolved with an increasing number of studies worldwide (Capps, 1910; Barsch, 1996; Haeberli, 2006; Berthling et al., 2011; Jones et al., 2019a; Janke and Bolch, 2021). In the last
decade, the identification and compilation of rock glacier inventories has sparked heated debate due to the intricate nature of these landforms (Berthling, 2011; Brardinoni et al., 2019). In response to the challenge posed by inconsistencies in the identification and compilation of rock glaciers, the IPA Action Group Rock Glacier Inventories and Kinematics (RGIK) was established in 2018 with the aim of developing widely accepted guidelines for rock glacier inventorying, thereby fostering a globally consistent and comprehensive approach to rock glacier inventories (Delaloye et al., 2018; RGIK, 2023). Through the
efforts of RGIK, the baseline and practical guidelines have been documented and updated in several versions, which greatly promote the global assemblage and uniform completion of rock glacier inventories (RGIK, 2023). This paper closely follows the RGIK guidelines in the conceptual definition of rock glaciers.

Rock glaciers are important to map and monitor for several reasons. First, they serve as visible indicators of frozen ground and provide essential information about the presence and extent of mountain permafrost (Barsch, 1996; Haeberli, 2006).
Therefore, they are valuable for assessing permafrost distributions (Boeckli et al., 2012; Schmid et al., 2015; Hassan et al., 2021; Li et al., 2024). Previous studies have used rock glacier inventories to assess permafrost maps in different regions. For instance, Boeckli et al. (2012) developed an Alpine Permafrost Index Map for the European Alps by calibrating a statistical model with rock glacier inventories. Schmid et al. (2015) used rock glaciers mapped from Google Earth to validate permafrost maps in the Hindu Kush Himalayan region. Similarly, Hassan et al. (2021) and Li et al. (2024) used rock glacier inventories
to model the permafrost probability distribution in their study areas. Second, rock glaciers are an integral component of mountain hydrological system, especially in arid regions (Corte, 1976; Azócar and Brenning, 2010; Rangecroft et al., 2013, 2015; Munroe, 2018), which is a potentially significant water resource that remains poorly quantified. Jones et al. (2021a) estimated that $62.02 \pm 12.40$ Gt of water volume equivalent (WVEQ) is stored within rock glaciers globally. The ratio of rock glacier-to-glacier WVEQ was estimated to be 1:618, which is expected to further increase with the ongoing melting of glaciers
(Jones et al., 2021a). Given the arid conditions of much of the western Tibetan Plateau, the inventory of rock glaciers is critical in assessing potential water resources in these regions. Third, the kinematic behaviour of rock glaciers is sensitive to changes in permafrost temperature and pore-water pressure, which are influenced by climate forcing such as air temperature and precipitation (e.g., Arenson et al., 2002, 2005; Cicoira et al., 2019a,b). Numerous studies have demonstrated a decadal to multi-





decadal acceleration trend in rock glacier velocity in many regions such as the European Alps (e.g., Delaloye et al., 2010; Marcer et al., 2021), Northern Tien Shan (Kääb et al., 2021) and the Andes (Vivero et al., 2021). Based on these global trends, Rock Glacier Velocity (RGV) has been added as a new product of the Essential Climate Variable (ECV) Permafrost by the Global Climate Observing System (Zemp et al., 2022). Fourth, rapid movement or destabilization of rock glaciers can trigger geohazards such as rockfalls, debris flows, and lake outbursts, posing a potential risk to nearby human infrastructure and facilities (Janke and Bolch, 2021; Marcer et al., 2021).

A full understanding of the role of rock glaciers on permafrost distribution, mountain hydrology and hazards in regions such as the Tibetan Plateau is currently hampered by the lack of comprehensive and systematic inventories. Compiling a comprehensive inventory constitutes the first step towards monitoring the long-term evolution of rock glaciers and understanding the changes of mountain permafrost under climate change. In recent years, rock glacier inventories in several local areas on the Tibetan Plateau have been established by visually interpreting optical images of different sources, and in some cases, Interferometric Synthetic Aperture Radar (InSAR) maps (Jones et al., 2018, 2021b; Ran and Liu, 2018; Hassan et al., 2021; Reinosch et al., 2021; Cai et al., 2021; Zhang et al., 2021; Bolch et al., 2022; Zhang et al., 2022; Hu et al., 2023; Zhang et al., 2023; Li et al., 2024) (see Table S1). However, the coverage remains patchy, and a plateau-wide open-access inventory compiled from a consistent set of images using a systematic methodology is currently still lacking, hence the purpose of this study.

However, the production of a rock glacier inventory through visual interpretation requires strong geomorphological expertise, and is labour-intensive and time-consuming (Barsch, 1996; RGIK, 2023). Rock glaciers exhibit spectral properties similar to their surrounding environment, making it challenging to identify on optical remote sensing images (Robson et al., 2020). Moreover, in high mountain environments, there are various landforms that resemble rock glaciers, such as debris-covered glaciers, rock avalanches, debris flows, and fluvial landforms (Haeberli et al., 2006; Robson et al., 2020). As a result, inexperienced analysts are prone to making erroneous judgments. With the development of artificial intelligence, deep learning models have become valuable tools for mapping complex landforms such as rock glaciers. Deep learning models are able to learn the visual patterns of objects and to identify features in previously unseen images with high accuracy (LeCun et al., 2015; Huang et al., 2020). In recent years, several studies have successfully employed deep learning techniques for the automatic detection of rock glaciers, yielding satisfactory results (Feng et al., 2019; Robson et al., 2020; Marcer, 2020; Xu et al., 2021; Erharter et al., 2022; Hu et al., 2023). However, the methods employed in previous studies are not systematic over large areas, leading to inconsistencies and patchy coverage.

In this study, we created the first region-wide inventory of rock glaciers on the Tibetan Plateau, i.e., TPRoGI [v1.0], using a deep learning method based on DeepLabv3+ model. It is expected that the benchmark dataset produced by this study will be maintained and updated in the future and will facilitate the investigation into many scientific questions related to rock glaciers and mountain permafrost on the Tibetan Plateau.

## 2 Study area

The Tibetan Plateau is part of High Mountain Asia, covering an area of approximately 2.5 million km$^2$ with an average elevation of over 4,500 m above sea level (Royden et al., 2008). The Tibetan Plateau has a continental climate dominated by the Indian monsoon, the East Asia monsoon, and the Westerlies. The monsoon brings warm and moist air in summer, while

the Westerlies bring dry and cold air in winter. The interaction between monsoons and Westerlies causes distinct seasonal climate variations and significant diurnal temperature differences on the Tibetan Plateau (Yao et al., 2012). Due to its high altitude and extreme weather conditions, the Tibetan Plateau has the largest cryosphere extent outside the Arctic and Antarctic regions and the largest area of permafrost terrain in the mid- and low-latitude regions (Zou et al., 2017).

Bolch et al. (2019b) split High Mountain Asia into 22 subregions based on their topographical and climatological

characteristics, of which 13 were situated in the Tibetan Plateau. We selected all the 13 subregions as study areas for this work, thus covering most of the Tibetan Plateau (Fig. 1), as well as the Qaidam basin, which was not a subregion in Bolch et al. (2019b)'s study. The Hindu-Kush Himalayan region was excluded from this study due to its distinct topo-climatic environment, requiring more training data to capture the variability in rock glacier types, but is a subject of future work.


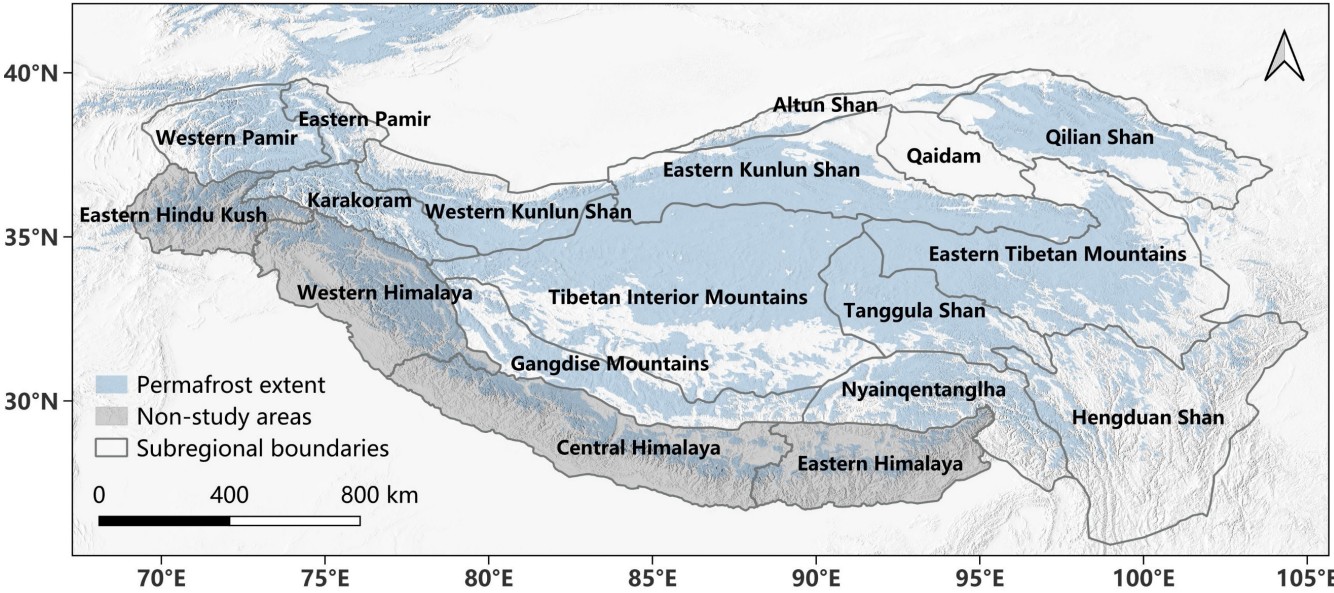

**Figure 1.** Study area (the Tibetan Plateau). The Hindu-Kush Himalayan region is excluded from this study. The permafrost extent map is from Obu et al. (2018).



## 3 Data

### 3.1 Planet Basemaps

We used a large volume of optical imagery from high-resolution satellite data, i.e., Planet Basemaps, as training images. Planet Labs Inc. generates the Basemaps product using imagery and data from its fleet with over 200 earth imaging satellites (Nass et al., 2019). The three-band (red, green, blue) imagery contains well-processed, scientifically accurate, and analyses-ready mosaics with a 4.77 m spatial resolution, visual consistency, and cloud mitigation (Nass et al., 2019). The visual

consistency of Planet Basemaps is crucial for developing a comprehensive map of rock glaciers over broad regions. Furthermore, we chose images from a single sensor to ensure consistent quality and time stamp of the source images. To train the deep learning model and infer new rock glaciers, we mostly utilized the Planet Basemaps mosaics from the third quarter (July-September 2021) supplemented with images from the fourth quarter (October-December 2021) when needed to mitigate image quality problems in the third-quarter images, such as shadows and image distortion.

### 3.2 Existing rock glacier local inventories for training

To create a set of robust and diverse training data, we compiled existing rock glacier local inventories from multiple regions. Utilizing a multi-source approach helps increase the volume and diversity of the training dataset while mitigating the subjectivity and possible biases introduced by individual inventories. To incorporate more high-quality data, we included rock glaciers not only from the Tibetan Plateau but also from other regions, with a total of six local inventories comprising both

intact and relict rock glaciers (Table 1).





**Table 1.** Information of rock glacier local inventories selected for training deep learning model.

| Location | Number of rock glaciers | Number of intact rock glaciers | Number of relict rock glaciers | Image sources | Method | Reference |
|---|---|---|---|---|---|---|
| Western Kunlun Shan | 413 | 413 | 0 | ALOS-1 PALSAR-1, Sentinel-2, Google Earth | InSAR, deep learning, visual analysis | Hu et al. (2023) |
| Hunza River Basin | 616 | 450 | 166 | Google Earth | Visual analysis | Hassan et al. (2021) |
| Poiqu River Basin | 370 | 370 | 0 | Pléiades, Google Earth | Visual analysis | Bolch et al. (2022) |
| Daxue Shan | 295 | Unknown | Unknown | Google Earth | Visual analysis | Ran and Liu (2018) |
| Northern Tien Shan | 551 | Unknown | Unknown | ERS-1/2 tandem mission, ALOS-1 PALSAR-1, ALOS-2 PALSAR-2, Sentinel-1, Google Earth, Bing Maps | InSAR, visual analysis | Kääb et al. (2021) |
| French Alps | 3,281 | 1,498 | 1,783 | IGN ortho-imagery | Visual analysis | Marcer et al. (2017) |

Prior to generating the final training dataset, we performed a quality control to account for the various source images and compilation strategies employed among these inventories. As a result, we manually checked and modified rock glacier boundaries by overlaying and visually checking the previously inventoried rock glaciers on our Planet Basemaps images. For 150 example, rock glaciers that were difficult to recognize at places where the image quality was poor or covered by shadows were removed; when we identified missing rock glaciers in previous inventories, these were manually added. Since the front is a critical feature of a rock glacier, we followed the RGIK guidelines to use the extended geomorphological footprints to delineate rock glacier training samples (RGIK, 2023). We finally collected 4,085 rock glacier polygons as training samples.

### 3.3 Topo-climatic datasets

To analyse the patterns of rock glacier distribution and the associated environmental factors, we used several topo-climatic datasets including (1) the 30-m-resolution National Aeronautics and Space Administration Digital Elevation Model (NASADEM) (Crippen et al., 2016), (2) the 0.1°×0.1° monthly mean annual air temperature (MAAT) data from January 1982 to the present derived from the Noah 3.6.1 model in the Famine Early Warning Systems Network (FEWS NET) Land Data Assimilation System (FLDAS) (McNally et al., 2018), (3) the mean annual ground temperature (MAGT) data from 2000 to 160 2016 at 1-km spatial resolution produced by Obu et al. (2018), and (4) the 0.1°×0.1° monthly precipitation data from 2001 to 2020 from the Integrated Multi-satellitE Retrievals for GPM (IMERG) (Huffman et al., 2019).



### 3.4 Auxiliary data

We also incorporated additional data sources including Google Earth images, ESRI basemaps, and the information on glacier and permafrost distributions. Google Earth images and ESRI basemaps were used as supplementary data to aid in the

identification and validation of rock glaciers by using high-resolution images (Yu and Gong, 2012). For the glacier and debris-covered glacier data, we utilized the widely recognized Randolph Glacier Inventory (RGI v6.0), which provides global coverage of glacier outlines (Pfeffer et al., 2014). The RGI offers a valuable reference for distinguishing rock glaciers from adjacent glaciers. Regarding permafrost extent, we relied on the map for the northern hemisphere produced by Obu et al. (2018).

## 4 Methodology

### 4.1 Deep learning-based rock glacier mapping approach

We propose a systematic deep learning-based approach for mapping rock glaciers on the Tibetan Plateau. The workflow of the mapping approach is illustrated in Fig. 2. The mapping process comprises two primary stages: (i) deep learning mapping and (ii) manual improvement, which will be elaborated in the following subsections.


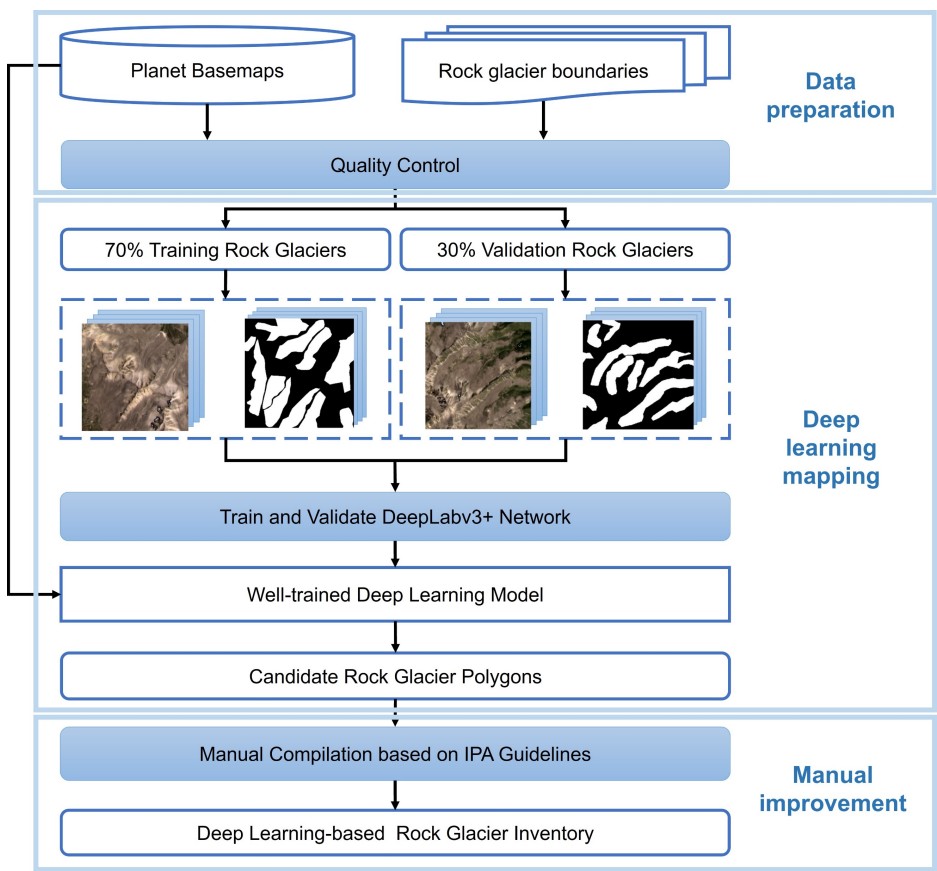

**Figure 2.** Flowchart of the deep learning-based approach for mapping rock glaciers.

### 4.1.1 Deep learning mapping

DeepLabv3+, introduced by Chen et al. (2018), was selected as the neural network architecture for the deep learning model, with Xception71 serving as its backbone (Chollet, 2017). DeepLabv3+ is specifically designed for semantic segmentation tasks and has been proven to excel in mapping permafrost landforms (Huang et al., 2020; Hu et al., 2023). Xception71 is a convolutional neural network architecture consisting of 71 layers and encompasses approximately 42 million parameters (Chollet, 2017).

Our deep learning model takes a three-channel image with red, green, and blue (RGB) bands as input and outputs a binary image indicating the occurrence of rock glaciers. The topographic information such as slope or elevation was not used because this model only accepts three image bands as input. For the model training, 70% of rock glacier boundaries from the six local inventories were extracted, with the remaining 30% kept for validation. To incorporate context information from the surrounding area of a rock glacier, we established a buffer area of 1,500 meters and extracted a subset of Planet images. These



images were then subdivided into patches of 480 × 480 pixels with an overlap of 160 pixels. The binary label patches were
created by rasterizing rock glacier polygons (Huang et al., 2020).

During the training process, the model was validated simultaneously using the subset of 30% rock glacier boundaries.
The intersection over union (IoU) was employed as the accuracy metric of validation, which is defined as:

$$\text{IOU}(A, B) = area(A \cap B)/area(A \cup B) ,\qquad\qquad (1)$$

where $A$ denotes the mapped polygon and $B$ is the reference polygon. The IoU scores range from 0 to 1, and a higher value
indicates a higher accuracy (Huang et al., 2020).

Once trained, the deep learning model was validated using images across the entire Tibetan plateau. We calculated the
areas of true positives (TP), false positives (FP), and false negatives (FN), and then calculated the precision, recall, and F1
score using the following equations (Huang et al., 2020):

$$Precision = TP/(TP + FP) ,\qquad\qquad (2)$$

$$Recall = TP/(TP + FN) ,\qquad\qquad (3)$$

$$F1 = 2 \times Precision \times Recall/(Precision + Recall).\qquad\qquad (4)$$

Since the predicted polygons are subject to uncertainties due to varying qualities of imagery, training inventories, and
model accuracy, initial results are referred to as "candidate rock glacier polygons". These polygons were not considered
definitive rock glacier inventories but rather served as preliminary detection of rock glaciers, along with the locations and
boundaries, which were then refined as described below.

### 4.1.2 Manual improvement and independent validation

To ensure the accuracy and reliability of the dataset, a manual checking and improving process was carried out on the
candidate rock glacier polygons. By utilizing these polygons as a starting point, the subsequent manual compilation efforts
were significantly streamlined. The manual improvement process followed the standard guidelines recommended by the IPA
Action Group RGIK (RGIK, 2023). According to these guidelines, the mapped rock glaciers were visually checked based on
specific geomorphological features, notably the visible accumulation of talus material at the front and the presence of a lateral
extension of this talus material along the sides of the rock glacier. Additionally, certain rock glaciers may exhibit noticeable
convex-downslope or longitudinal-surface undulations, creating a ridge-and-furrow topography. We considered the extended
footprints of rock glaciers while restricting the horizontal distance between the upper front edge and the frontal talus base
within 50 m to exclude the possible exaggerated front. Following the global glacier inventory standards and given the resolution
limitations of Planet Basemaps (4.77 m), rock glaciers smaller than 10,000 m$^2$ (0.01 km$^2$) were excluded from the inventory
(RGIK, 2023).

We proposed four "R" operations to manually check the rock glacier candidate polygons:





    – "Remain": no operation if the polygon accurately outlines the rock glacier

– "Remove": remove the polygon if it is not a rock glacier

    – "Refine": modify the polygon if it was correctly identified as a rock glacier but the boundaries were not correctly outlined

    – "Retrieve": add a missing rock glacier and outline its boundary

The "Remove" operation is designed to exclude landforms that have been incorrectly identified as rock glaciers by the
deep learning model. These misidentified landforms commonly include debris-covered glaciers and rock avalanches. Debris-covered glaciers are glaciers that are partially covered by variable layers of debris (from a few centimeters to two meters) and are characterized by supraglacial features such as thermokarst features, supraglacial lakes, streams, and ice cliffs (Jones et al., 2019b; Racoviteanu et al. 2022; RGIK, 2023). Outlines from the RGI v6.0 inventory were used to visually remove polygons overlapping debris-covered glaciers. Rock avalanches, on the other hand, are composed of fragmented rocks that flow downhill
following large rock slope failures (Hungr et al., 2014). Unlike rock glaciers, rock avalanches typically lack any discernible pattern or order on the surface. The "Refine" operation involved the manual editing of the deep learning predicted rock glacier outlines to ensure that the polygon boundaries closely matched the observed boundaries of rock glaciers in the images. The "Retrieve" operation serves the purpose of adding missing rock glaciers to the inventory. Some rock glaciers can be overlooked by the deep learning model, either due to their subtle features or low-quality image data. Furthermore, in high mountain
environments, the convergence of multiple rock glacier units into a complex system is a frequent occurrence (RGIK, 2023). However, the deep learning model often tends to predict this system as a singular rock glacier. To anticipate such issues, we manually separated the system into smaller rock glacier units if their lateral boundaries were clearly observed in Planet Basemaps images.

Our team consisted of seven mappers and two independent reviewers. Each candidate rock glacier polygon was manually
examined and refined by visual interpretation of Planet Basemaps images following the four "R" operations by each mapper. In cases where the features of rock glaciers were uncertain and not clearly observable in Planet images, high-resolution Google Earth images and ESRI basemaps were utilized for more accurate visual inspection and analysis. An extended footprint of each rock glacier was yielded, from which we generated the primary marker, which is a point identifying a unique rock glacier unit or system (RGIK, 2023).

We proceeded with an independent validation process to assess the quality of the revised inventory. Given the difficulty in accurately evaluating the delineated boundaries, our validation focused primarily on verifying the primary markers. To conduct this validation, we randomly selected 2,110 samples (approximately 5% of the primary markers). Two independent reviewers examined all the selected samples using Google Earth images. Based on their independent assessments, each reviewer provided one of four decisions: "yes," indicating that the rock glaciers were correctly identified; "no," suggesting an
incorrect identification; "uncertain," denoting a lack of certainty in the identification; and "undefined," used when the examined





rock glaciers could not be clearly observed due to factors such as heavy snow cover, shadows, or the unavailability of high-quality images.

## 4.2 Adding attributes of the final revised rock glaciers

According to the IPA RGIK guidelines, there are three mandatory attributes for a rock glacier unit (RGU): the primary
ID, the associated rock glacier system (RGS), and the metadata (RGIK, 2023). In our inventory, the attribute ID is equivalent to the primary ID, which is formed by combining "RGU" with the WGS84 coordinates of the rock glacier, expressed in decimal degrees with four digits (RGIK, 2023). We were unable to provide the RGS information in our current inventory due to image resolution limitations and instance segmentation issues. We have included the metadata attribute, which contains information of source data, date of mapping, mapper's name, reviewer's name, and additional information, which are separately stored in
SOUR_DATA, MAP_DATE, MAPPER, REVIEWER, and ADDI_INF attributes (RGIK, 2023). The ADDI_INF provides information on whether the rock glacier has been recognized as a false identification by the reviewers. Furthermore, we computed the geomorphic and climatic attributes of each inventoried rock glacier to analyse their spatial distribution characteristics and the associated topo-climatic conditions. We derived the rock glacier area based on the polygon extent. The NASADEM was used to calculate the elevation, slope, and aspect of the rock glaciers (Crippen et al., 2016). The climatic
information, including MAAT, MAGT, and annual precipitation, of each rock glacier was extracted from the climatic data. We also calculated the annual potential incoming solar radiation (PISR) using the model described by Kumar et al. (1997). Table 2 lists all the attributes of the inventory.






**Table 2.** Attribute data dictionary for Tibetan Plateau rock glacier inventory shapefile.

| Attribute name | Description | Units |
|---|---|---|
| ID[1] | Rock glacier ID | |
| SOUR_DATA[2] | Source data | |
| MAP_DATE[2] | Date of mapping | |
| MAPPER[2] | Mapper's name | |
| REVIEWER[2] | Reviewer's name | |
| ADDI_INF[2] | Additional information | |
| LAT | Latitude | Degrees |
| LON | Longitude | Degrees |
| SUBREGION | Subregion of rock glacier | |
| AREA | Rock glacier area | $m^2$ |
| ELE_MEAN[3] | Mean elevation of rock glacier | m |
| ELE_MEDIAN[3] | Median elevation of rock glacier | m |
| ELE_MIN[3] | Minimum elevation of rock glacier | m |
| ELE_MAX[3] | Maximum elevation of rock glacier | m |
| SLO_MEAN[3] | Mean slope of rock glacier | Degrees |
| SLO_MEDIAN[3] | Median slope of rock glacier | Degrees |
| SLO_MIN[3] | Minimum slope of rock glacier | Degrees |
| SLO_MAX[3] | Maximum slope of rock glacier | Degrees |
| ASPECT[3] | Aspect of rock glacier | Degrees |
| MAAT[4] | Mean annual air temperature | °C |
| MAGT[5] | Mean annual ground temperature | °C |
| AP[6] | Annual precipitation | mm |
| PISR[3] | Annual potential incoming solar radiation | $kWh/m^2$ |

[1]ID is identical to the Primary ID attribute in the IPA RGIK guidelines.
[2]SOUR_DATA, MAP_DATE, MAPPER, REVIEWER, and ADDI_INF contain the information of Metadata attribute in the IPA RGIK guidelines.
[3]ELE_MEAN, ELE_MEDIAN, ELE_MIN, ELE_MAX, SLO_MEAN, SLO_MEDIAN, SLO_MIN, SLO_MAX, ASPECT, and PISR are attributed based on the 30-m-resolution National Aeronautics and Space Administration Digital Elevation Model (NASADEM) (Crippen et al., 2016) (https://search.earthdata.nasa.gov/search).
[4]MAAT is attributed based on the 0.1°×0.1° monthly mean annual air temperature (MAAT) data from January 1982 to the present derived from the Noah 3.6.1 model in the Famine Early Warning Systems Network (FEWS NET) Land Data Assimilation System (FLDAS) (McNally et al., 2018) (https://disc.gsfc.nasa.gov/datasets/FLDAS_NOAH01_C_GL_M_001/summary?keywords=MERRA-2%20and%20CHIRPS).
[5]MAGT is attributed based on mean annual ground temperature (MAGT) data from 2000 to 2016 at 1-km spatial resolution produced by Obu et al. (2018) (https://apgc.awi.de/dataset/pex).
[6]AP is attributed based on the the 0.1°×0.1° monthly precipitation data from 2001 to 2020 from the Integrated Multi-satellitE Retrievals for GPM (IMERG) (Huffman et al., 2019) (https://disc.gsfc.nasa.gov/datasets/GPM_3IMERGM_06/summary?keywords=GPM%20IMERG%20Final%20Precipitation%20L3%201%20month%20 0.1%20degree%20x%200.1%20degree%20V06%20(GPM_3IMERGM)).

## 4.3 Spatial analysis of rock glaciers

To investigate the spatial distribution characteristics of rock glaciers on the Tibetan Plateau, we conducted statistical analyses of their geomorphic features within a 50 km × 50 km grid cell. In each cell, we counted the number of rock glaciers



and calculated the average values for their areas, minimum elevations, and slopes. We also analysed the distribution patterns of their aspects in different subregions.

## 5  Results

Across the entire study area, the deep learning model predicted a total of 48,767 candidate rock glacier polygons (Fig. S1). After the manual improvement (cf. section 4.1.2), we produced an inventory consisting of 44,273 rock glaciers (Fig. 5 and further described in Sections 5.2 and 5.3). Below we first present the validation of our results from three perspectives: (i) validation of the deep learning model based on training and validation datasets (section 5.1.1); (ii) validation of the deep learning predicted rock glacier outlines based on manually improved rock glaciers used as our ground truth (section 5.1.1); 295    (iii) independent validation of inventoried rock glaciers based on visual examination (section 5.1.2).

### 5.1  Performance of deep learning-based rock glacier mapping approach

### 5.1.1  Deep learning model performance and output

Fig. 3a shows the IoU scores achieved by the deep learning model during the training and validating processes. Initially, both the training and validation IoU scores exhibit an upward trend, followed by a gradual stabilization. By the last iteration, 300    the model achieved an IoU score of 0.76 on the training dataset and 0.70 on the validation dataset, indicating that the model learned effectively from the training data and generalized well to the validation data.

To further evaluate the model performance, we applied the well-trained model to predict the rock glacier boundaries on both the training and validation datasets. The deep learning model accurately captured rock glacier characteristics within the training dataset, as evidenced by the close alignment between the predicted boundaries and the training polygons (Fig. 3b). 305    Fig. 3c further confirms that the model could generalize well to new datasets, with good agreement between predicted boundaries and validation polygons. However, difficulties in mapping rooting regions led to misalignment in those areas (Brardinoni et al., 2019).



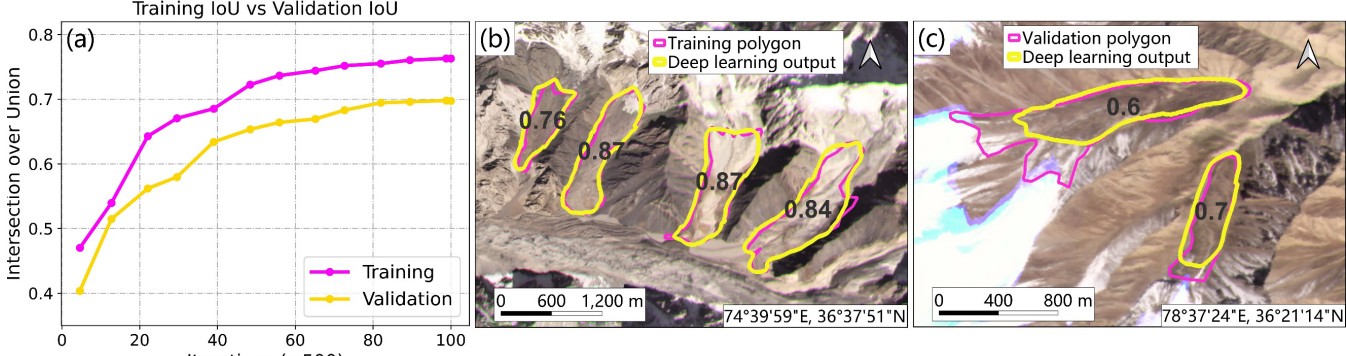


**Figure 3.** (a) IoU scores during the training and validation processes. Examples of the candidate rock glaciers are shown in (b) training and (c) validation regions using the well-trained deep learning model. The IoU scores are labelled on the mapped rock glaciers.

Table 3 presents the calculated recall, precision, and F1 score of the deep learning mapped polygons for each subregion, as well as for the entire study area. Over the entire study area, the F1 score was 0.63, which we consider satisfactory for rock glacier mapping. The highest performance was for the Hengduan Shan (F1 = 0.76), with F1 scores of the Eastern Pamir, Karakoram, Nyainqêntanglha, Western Kunlun Shan, Western Pamir, and Hengduan Shan subregions above 0.6, lower F1 scores for the Altun Shan, Eastern Kunlun Shan, Eastern Tibetan Mountains, Gangdise Mountains (0.27 – 0.36), and the lowest for Tibetan Interior Mountains subregion (0.16) (Table 3). This disparity arises from the scarcity of rock glaciers in certain

subregions, where the deep learning model generated a large number of falsely detected polygons and subsequently produced high false positives. Recall scores are generally higher than the precision scores, indicating that the false positives outweigh the false negatives in the model predictions. This finding suggests that the deep learning model possesses a strong capability for detecting rock glaciers.

However, deep learning alone also generates numerous falsely detected polygons, highlighting the need for manual

improvement. For example, Fig. 4 demonstrates the performance of the well-trained deep learning model in detecting and delineating rock glaciers in a new area - the Western Pamir, which was not included in the training process. As shown in Fig. 4a-c, there is good agreement between the deep learning output and the manually revised boundaries for a significant proportion of the rock glaciers in this area. However, Fig. 4 also illustrates some uncertainties associated with inaccurate boundary delineation, false detections, and missing identifications. For instance, as shown in Fig. 4d, a debris-covered glacier was falsely

identified as a rock glacier, while Fig. 4e highlights several missing rock glaciers, possibly due to their poorly developed geomorphological features.






**Table 3.** Performance of deep learning mapped polygons in different subregions.

| Subregion | Number of deep learning mapped polygons | Number of manually revised rock glaciers | TP[1] (km²) | FP[1] (km²) | FN[1] (km²) | Precision | Recall | F1 score |
|---|---|---|---|---|---|---|---|---|
| Altun Shan | 82 | 32 | 3.55 | 9.03 | 3.45 | 0.28 | 0.51 | 0.36 |
| Eastern Kunlun Shan | 517 | 180 | 22.81 | 60.57 | 33.46 | 0.27 | 0.41 | 0.33 |
| Eastern Pamir | 1,060 | 1,330 | 230.76 | 115.40 | 143.12 | 0.67 | 0.62 | 0.64 |
| Eastern Tibetan Mountains | 2,569 | 1,095 | 43.97 | 200.09 | 32.34 | 0.18 | 0.58 | 0.27 |
| Gangdise Mountains | 1,572 | 816 | 49.91 | 128.50 | 66.82 | 0.28 | 0.43 | 0.34 |
| Karakoram | 2,873 | 2,612 | 415.89 | 344.57 | 133.41 | 0.55 | 0.76 | 0.64 |
| Nyainqêntanglha | 14,161 | 16,222 | 1,095.42 | 876.49 | 465.46 | 0.56 | 0.70 | 0.62 |
| Qilian Shan | 1,367 | 1,047 | 77.21 | 129.90 | 68.68 | 0.37 | 0.53 | 0.44 |
| Tibetan Interior Mountains | 1,158 | 150 | 15.71 | 130.23 | 35.13 | 0.11 | 0.31 | 0.16 |
| Western Kunlun Shan | 779 | 1,019 | 116.44 | 69.40 | 87.56 | 0.63 | 0.57 | 0.60 |
| Western Pamir | 4,989 | 4,957 | 685.50 | 549.56 | 266.32 | 0.56 | 0.72 | 0.63 |
| Tanggula Shan | 4,010 | 2,402 | 166.92 | 288.34 | 61.34 | 0.37 | 0.73 | 0.49 |
| Hengduan Shan | 13,387 | 12,411 | 1,478.55 | 678.96 | 268.31 | 0.69 | 0.85 | 0.76 |
| Qaidam | 243 | 0 | 0 | 15.95 | 0 | 0 | N/A | N/A |
| Entire study area | 48,767 | 44,273 | 4,403.43 | 3,596.49 | 1,664.61 | 0.55 | 0.73 | 0.63 |

[1]TP (true positive), FP (false positive) and FN (false negative) are expressed as the total areas.

**Figure 4.** (a) An example area in Western Pamir showing the deep learning outputs (in red) and manually revised rock glacier boundaries (in green). Clean and debris-covered glacier extents (light blue) are from the Randolph Glacier Inventory (RGI v.6) (Pfeffer et al., 2014); (b-c) enlarged views of the areas showing good agreement betweendeep learning outputs and revised boundaries; (d) enlarged view showing a false detection example in the center; (e) enlarged view showing multiple missing identifications.



### 5.1.2 Independent validation of the inventoried rock glaciers

Results of the independent review based on the 2110 rock glacier primary markers are presented in Table 4 and show that approximately 87% of the primary markers were assigned as correctly identifying rock glaciers by both reviewers. This

indicates that most of the sampled features met the criteria and characteristics of rock glaciers. Additionally, the evaluation process identified that only approximately 1% and 6% of the primary markers were assigned as false identifications by the two reviewers, respectively. This signifies that the occurrence of misclassifications or false positives within the inventory is relatively low (below 10%). The discrepancy in the "no" decision numbers between the two reviewers can be attributed to the differences in the operators' judgments (Brardinoni et al., 2019).


**Table 4.** Independent validation results of sampled Tibetan Plateau rock glacier inventory (n = 2110 samples).

| Reviewer | Number of "yes" | Number of "no" | Number of "uncertain" | Number of "undefined" |
|---|---|---|---|---|
| Reviewer 1 | 1836 | 17 | 42 | 215 |
| Reviewer 2 | 1844 | 127 | 44 | 95 |

### 5.2 Rock glacier inventory on the Tibetan Plateau: TPRoGI [v1.0]

After manual improvement, our plateau-wide inventory encompasses 44,273 rock glaciers, including both intact and relict

types (Fig. 5). The inventoried rock glaciers cover a total area of approximately 6,000 km$^2$ (6,068,043,348 m$^2$). The mean area is 0.14 km$^2$. The largest rock glacier occupies 4.6 km$^2$, whereas most of them (90.6%) are smaller than 0.3 km$^2$ (Fig. 6a). In terms of elevation, most rock glaciers (95.0%) exhibit minimum elevations between 4,000 m and 5,500 m above sea level (m.a.s.l.), with an average value of 4,729 m.a.s.l. (Fig. 6b). The highest rock glacier is situated at an elevation of 5,839 m.a.s.l. in the Tibetan Interior Mountains, whereas the lowest lies at 2,717 m.a.s.l. in Western Pamir. Rock glaciers develop on slopes

with varying gradients, and approximately 90% of them occur on slopes between 10° to 25° with an average slope angle of 17.7° (Fig. 6c). Also, the compiled rock glaciers are distributed across various slope orientations with preferences at the north- and west-facing slopes (Fig. 6d).

Rock glaciers predominantly occur in cold environments with temperatures at or slightly below freezing. A significant proportion of rock glaciers (66.3%) thrive in areas where the MAAT ranges between -5 °C and 0 °C (Fig. 6e). Furthermore,

71.7% of the rock glaciers exhibit MAGT between -5 °C and 0 °C (Fig. 6f). On average, the MAAT and MAGT for these rock glaciers are -2.7 °C and -1.6 °C, respectively. Approximately 82% of the rock glaciers are situated in regions with annual precipitation ranging from 300 mm to 1,000 mm, with an average of 597 mm (Fig. 6g). About 85% of the rock glaciers receive incoming solar radiation (PISR) between 2,500 kWh/m$^2$ and 3,500 kWh/m$^2$ annually, with a mean value of 2,930 kWh/m$^2$ (Fig. 6h).

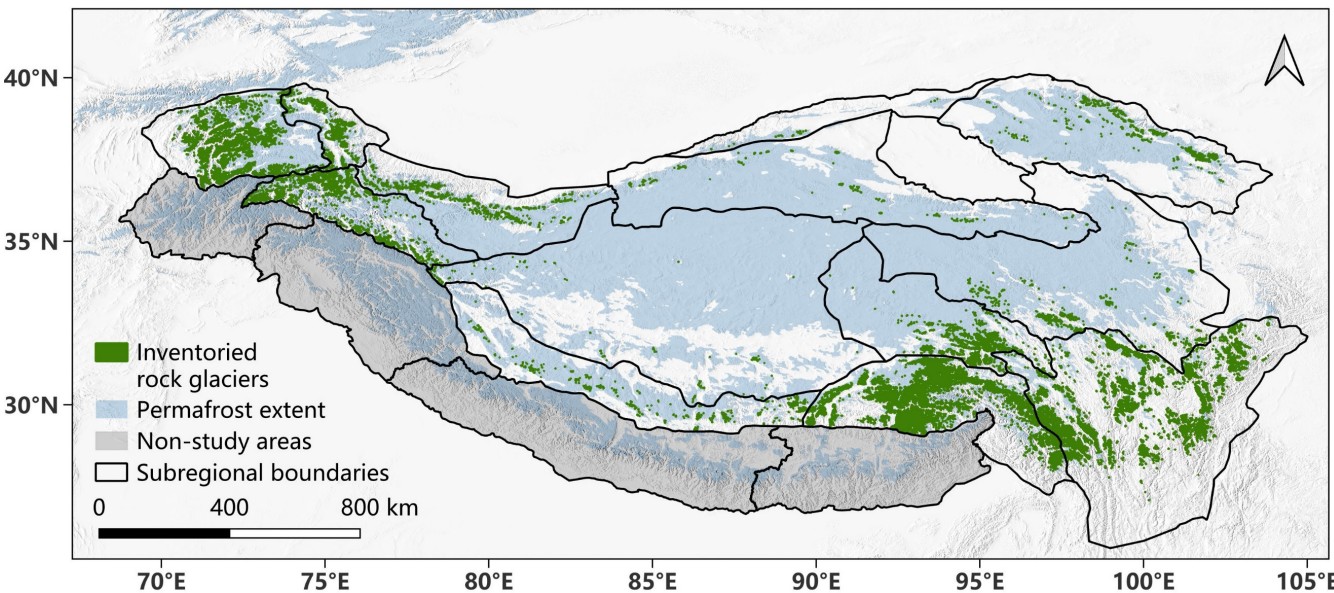


**Figure 5.** Rock glacier inventory on the Tibetan Plateau (TPRoGI). The permafrost in Hengduan Shan is overlapped by the rock glaciers thus not visible on the map. The permafrost extent map is from Obu et al. (2018).

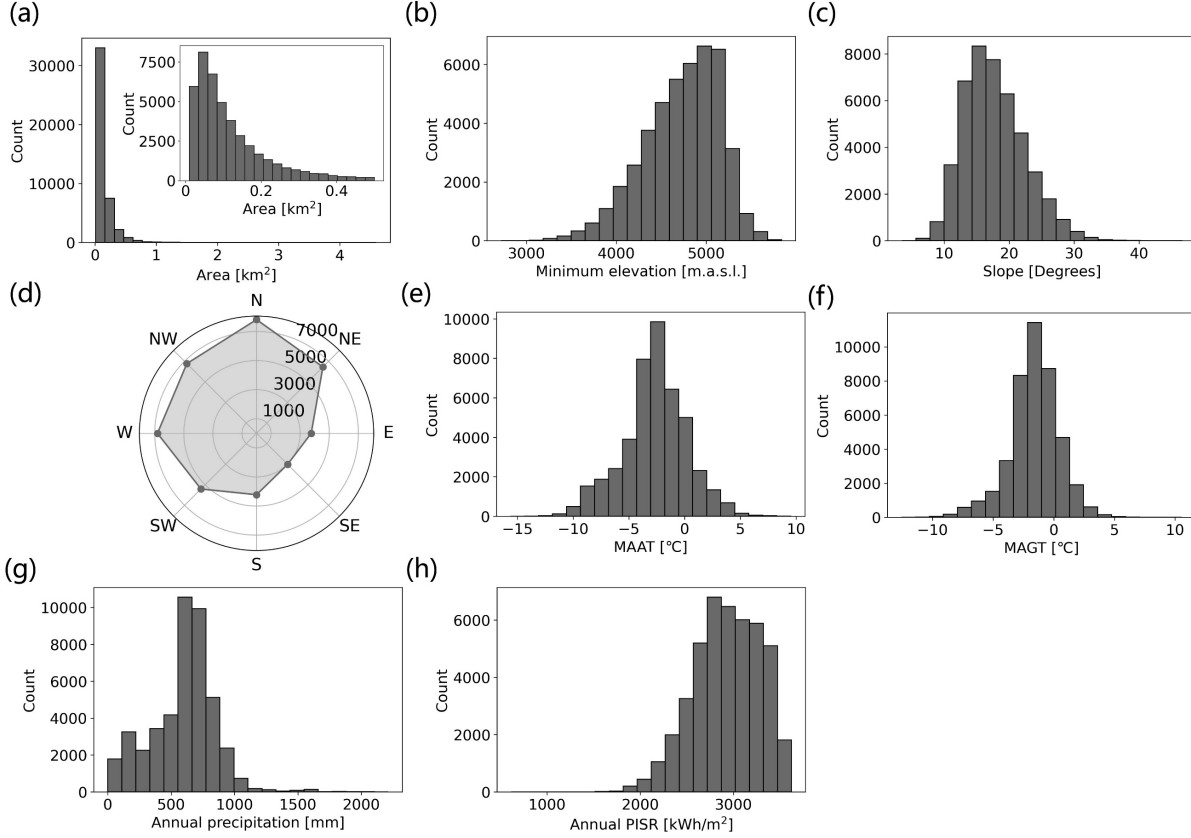

**Figure 6.** Statistical summaries of the geomorphic and current climatic features of rock glaciers in the study region. (a) The areal histogram of all the rock glaciers on the Tibetan Plateau. The inset shows the areas smaller than 0.5 km². (b)-(h) are histograms of the minimum elevations, slopes, aspects of the rock glaciers with the radial axis representing the counts, Mean Annual Air Temperature (MAAT), Mean Annual Ground Temperature (MAGT), annual precipitation, and annual Potential Incoming Solar Radiation (PISR), respectively.

### 5.3 Spatial distribution characteristics of rock glaciers

Fig. 7 presents the spatial distribution and geomorphic characteristics of rock glaciers on the Tibetan Plateau within 50 km grid cells. Rock glaciers are widespread in the northwestern and southeastern plateau and densely distributed in the Western Pamir and Nyainqêntanglha, while they are scarce in the inner plateau (Fig. 7a). No rock glacier was found in the Qaidam region, presumably due to the absence of permafrost and the occurrence of few mountains there. Rock glaciers in the western plateau have larger areas (mean = 0.21 km²) than in the eastern plateau (mean = 0.11 km²), as evident in Fig. 7b. Notably, a decreasing gradient is observed in minimum elevations of rock glaciers, with higher elevations in the Gangdise Mountains and lower elevations towards the east and west directions (Fig. 7c). The average slopes of rock glaciers are larger in northwestern Karakoram and southeastern plateau, suggesting a tendency for rock glaciers to develop on steeper slopes in these areas (Fig. 7d).





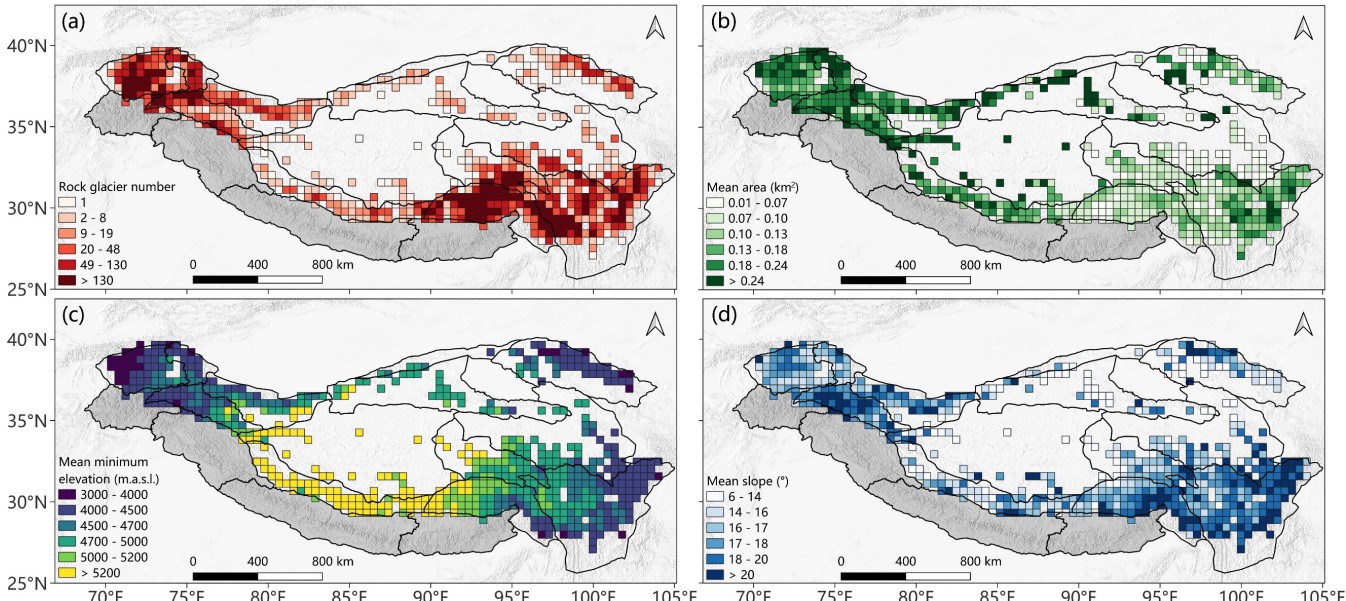

**Figure 7.** Rock glacier (a) density, (b) area, (c) minimum elevation and (d) slope averaged over grid cells of 50 km × 50 km.

Rock glacier aspects across different subregions are depicted in Fig. 8, characterized by a discernible west-east gradient and similarities between neighbouring subregions. Specifically, the ones found in the western plateau (Western Kunlun Shan, Karakoram, Eastern Pamir, Western Pamir) display no distinct preference towards any specific orientation, whereas those situated in the central part of the plateau (Altun Shan, Eastern Kunlun Shan, Tibetan Interior Mountains, Gangdise Mountains) primarily face north. Conversely, rock glaciers in the eastern plateau (Qilian Shan, Eastern Tibetan Mountains, Tanggula Shan, Hengduan Shan, Nyainqêntanglha) exhibit a prevalent preference for north and west orientations.



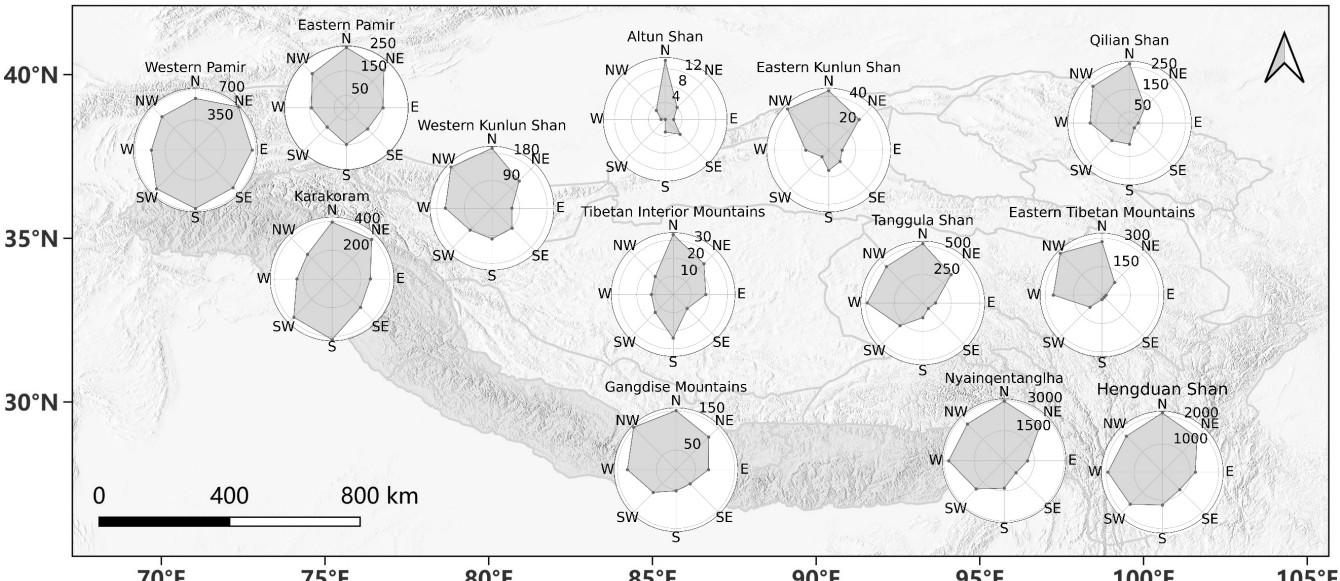

**Figure 8.** Rock glacier aspects in different subregions of the study area.

# 6 Discussion

## 6.1 Limitations of the deep learning-based mapping approach

### 6.1.1 Limitations of Planet Basemaps imagery

The quality of the source images plays a crucial role in the uncertainty of the predicted results as the deep learning model accuracy heavily relies on high-quality input images. However, rock glaciers are frequently found in regions characterized by poor image quality due to factors associated with cloud cover, shadows, and distortions, which are common in mountainous areas. These challenges have a substantial impact on the accuracy of predictions. Consequently, when the deep learning model is input with images suffering severe quality issues, it may fail to identify rock glaciers within that region (Fig. S5).

### 6.1.2 Limitations of the deep learning model

The mapped results generated by the deep learning model still have significant uncertainties associated with inaccurately predicted boundaries, false detections, and missing identifications (Fig. 4). Despite utilizing the powerful neural network DeepLabv3+ as the model structure, the training and validation IoU scores fall below 0.8 (Fig. 3). When applied to the entire study area, the uncertainty increases further, with a precision of 0.55, a recall of 0.73, and a F1 score of 0.63 (Table 3). These results are comparable to Robson et al. (2020)'s results, which obtained a precision of 63.9% to 68.9% and a recall of 75% to





75.4%. Both results highlighted the challenges of using deep learning to map rock glaciers fully automatically in high mountain
environment.

Furthermore, the learning performance of the model can be hindered by limited and biased training samples. Our training
samples were derived from six local inventories, encompassing 4,085 rock glacier polygons. Due to the limited quantity of our
training dataset, the model may struggle to fully capture the complexity and diversity of the training samples. Consequently,
its generalization ability and accuracy may be compromised when presented with unfamiliar images (Rice et al., 2020).
Additionally, the six local inventories were compiled by different operators from various institutes. The divergent knowledge
and expertise among these operators can introduce inconsistencies in judgments, resulting in subjectivity and bias within the
training dataset. As a result, inconsistent and biased training samples can potentially confuse the model, thereby impairing its
ability to accurately identify rock glaciers (Ren et al., 2018).

Additionally, it is important to note that the deep learning model can only map the areas of rock glaciers and is not capable
of performing instance segmentation, which would accurately segment individual rock glacier units (Erharter et al., 2022).
Consequently, the model tends to predict the entire rock glacier system, composed of several adjacent rock glacier units, as a
single entity.

### 6.1.3  Limitations of manual improvement

The manual examination and refinement were assigned by multiple individuals with varying levels of experience, which
inevitably introduced subjectivity, human errors, and potential inconsistencies (Brardinoni et al., 2019). Moreover, accurately
depicting the boundaries of rock glaciers via manual delineation can be challenging due to the 4.77 m resolution of the
interpreted images and thus the mapped rock glaciers inherently contain uncertainties (Jones et al., 2018).

Furthermore, delineating the upper and lateral boundaries within rock glacier systems presents even greater uncertainties
(Brardinoni et al., 2019). In comparison to the lower boundary in the front and lateral margin regions, the upper boundary in
the rooting zone and the lateral boundary between rock glaciers within a system often lack pronounced geomorphological
features and thereby require more precise interpretation of surface texture and colour variations. As a result, the delineation of
upper and lateral boundaries is inherently ambiguous and subjective (Schmid et al., 2015; Jones et al., 2018; Erharter et al.,
2022). Due to the difficulty in delineating lateral boundaries and the limitations imposed by image resolution, the separation
of rock glacier systems is uncertain. Therefore, some rock glacier systems, particularly the smaller ones lacking pronounced
features of lateral boundaries, may not be effectively separated (Fig. S4).

### 6.2  Comparison with existing local inventories

We compared the number of inventoried rock glaciers in our study with existing local inventories on the plateau, including
Daxue Shan (Ran and Liu, 2018; Cai et al., 2021), Nyainqêntanglha (Reinosch et al., 2021; Zhang et al., 2023; Li et al., 2024),
Hunza Basin (Hassan et al., 2021), Gangdise Mountains (Zhang et al., 2022), and West Kunlun Shan (Hu et al., 2023), as



shown in Table 5. The number of inventoried rock glaciers in our study is generally comparable to those in Daxue Shan and Hunza Basin. However, our inventory has more rock glaciers than the inventories in Gangdise Mountains and West Kunlun Shan, and fewer rock glaciers than the inventories in Nyainqêntanglha.

**Table 5.** Comparisons of the numbers of inventoried rock glaciers with existing local inventories.

| Location | Reference of existing local inventory | Number of inventoried rock glaciers in previous inventory | Number of inventoried rock glaciers in this study |
|---|---|---|---|
| Daxue Shan | Ran and Liu (2018) | 295 | 256 |
| Daxue Shan | Cai et al. (2021) | 344 | 256 |
| Western Nyainqêntanglha Range | Reinosch et al. (2021) | 1,433 | 798 |
| Hunza Basin | Hassan et al. (2021) | 616 | 647 |
| Gangdise Mountains | Zhang et al. (2022) | 132 | 816 |
| Western Kunlun Shan | Hu et al. (2023) | 413 | 2,145 |
| Nyainqêntanglha | Zhang et al. (2023) | 20,531 | 16,222 |
| Guokalariju | Li et al. (2024) | 5,057 | 4,000 |


    These discrepancies can be explained by inherent sources of error within each dataset. As highlighted in the RGIK guidelines (RGIK, 2023), operator judgment in compiling rock glacier inventories can vary over time, leading to discrepancies between inventories created at different time periods. Even within the same time frame, differences in operator experience can result in significant variations in judgment (Brardinoni et al., 2019). For example, the delineation of the upper boundary of

rock glaciers in rooting regions is challenging and can vary among different operators (Brardinoni et al., 2019). In the Hunza Basin, our delineated rock glaciers had lower upper boundaries compared to the results of Hassan et al. (2021) (Fig. S2). Additionally, small rock glaciers can be difficult to recognize due to the lack of distinct characteristics. In the Nyainqêntanglha region, some small landforms were included as rock glaciers in the inventories of Reinosch et al. (2021) and Li et al. (2024) but were excluded from our inventory (Fig. S3). Moreover, it is common in mountainous environments for several rock glacier

units to merge into a complex system (RGIK, 2023). Some operators may delineate this system as a single polygon, while others may separate it into smaller polygons. This can be observed in the case of Daxue Shan, where some systems were delineated as single polygons in our inventory but were separated into smaller polygons in the inventories of Ran and Liu (2018) and Cai et al. (2021) (Fig. S4).

    Another significant factor contributing to discrepancies in inventories is the use of different image sources. Images with

varying types, resolutions, and qualities can greatly influence the inventory results. The use of InSAR images, for example, is beneficial for detecting actively moving rock glaciers but may have poor performance in identifying slowly moving or relict rock glaciers (Liu et al., 2013; Hu et al., 2023). Moreover, images with low resolution used in some of the previous inventories





may not clearly reveal the morphological characteristics of rock glaciers, increasing the probability of missing identifications. In the Western Kunlun Shan region, our inventory compiled more rock glaciers by using Planet Basemaps images (4.77 m resolution) compared to Hu et al. (2023), whose inventory was based on Sentinel-2 images (10 m resolution). Additionally, images with quality issues caused by clouds, snow, shadows, and image distortion can lead to missed identifications of rock glaciers. In some areas of Nyainqêntanglha, for instance, some rock glaciers were obscured by clouds in Planet Basemaps images and were missed in our inventory, but they were visible in Google Earth images and had been included in the inventories of Reinosch et al. (2021) and Li et al. (2024) (Fig. S5). Since the discrepancies between inventories can arise from various sources, conducting further quantitative comparisons on the accuracies of rock glacier locations and boundaries poses challenges.

### 6.3 Significance of the inventory and future work

To our knowledge, the creation of the new inventory on the Tibetan Plateau represents the most extensive collection of rock glaciers published worldwide. This large dataset offers exciting prospects for advancing various research areas related to rock glaciers, including permafrost distribution, mountain hydrology, climate impacts on the permafrost environment, and geohazards as introduced in Section 1.

First, our new inventory enables more accurate assessments of permafrost distribution, allowing researchers to refine existing permafrost maps and enhance our understanding of permafrost characteristics on the Tibetan Plateau (Schmid et al., 2015; Hassan et al., 2021; Zou et al., 2017; Li et al., 2024). We underline that the lack of comprehensive rock glacier information on the plateau had previously limited permafrost assessment studies in this region. Cao et al. (2021) found that a model driven by rock glacier observations led to an overestimation of permafrost extent by about 8.4-13.1% on the Tibetan Plateau compared to a model using in situ measurements. Nevertheless, they used datasets from the Himalayan range as an alternative due to the limited availability of rock glacier observations on the plateau.

With respect to hydrology, Jones et al. (2021a) had estimated the global water contribution from rock glaciers and highlighted the lack of rock glacier data in certain regions, including the Tibetan Plateau. Our inventory fills the data gap in this critical region of High Mountain Asia, providing an opportunity to investigate the potential water storage available within rock glaciers (Corte, 1976; Azócar and Brenning, 2010; Jones et al., 2019a; Schaffer et al., 2019; Wagner et al., 2020, 2021) and the contribution of rock glacier meltwater to streamflow (Geiger et al., 2014; Wagner et al., 2016).

Moreover, our inventory serves as a guide for establishing rock glacier monitoring sites on the plateau, contributing to the study of the long-term evolution of rock glaciers and the impacts of climate change on mountain permafrost in this region. Systematic monitoring of rock glacier velocities has been established in the European Alps (e.g., Delaloye et al., 2010; Marcer et al., 2021), Northern Tien Shan (Kääb et al., 2021), and the Andes (Vivero et al., 2021). Currently no such monitoring sites exist on the Tibetan Plateau due to the lack of information on rock glacier distribution.





Lastly, the new inventory developed in this study will contribute to the evaluation of rock glacier hazards and risks,
providing important information for geohazard management and enabling informed decision-making regarding infrastructure
planning on the Tibetan Plateau (Hassan et al., 2021; Janke and Bolch, 2021).

This benchmark dataset will be maintained and updated in the future. Work is currently ongoing to evaluate the rock
glacier kinematics on the Tibetan Plateau based on the developed inventory as well as to validate the deep learning rock glacier
output for the Hindu Kush Himalaya regions. Additionally, this new inventory can serve as a benchmark dataset for training
new deep learning models.

## 7  Data availability

The rock glacier inventory for the Tibetan Plateau can be accessed at https://doi.org/10.5281/zenodo.10732042 (Sun et
al., 2024).

## 8  Conclusions

In this study, we proposed a deep learning-based approach for mapping rock glaciers and created the first plateau-wide
inventory i.e., TPRoGI [v1.0], encompassing 44,273 rock glaciers. This inventory fills the gap in the rock glacier data on the
Tibetan Plateau and provides a baseline dataset for monitoring mountain permafrost in this region. Findings from the current
study are summarized as follows: (1) the deep learning model demonstrates a promising capability in detecting and outlining
rock glaciers and can serve as a valuable tool for inventorying rock glaciers across large regions; (2) rock glaciers are
widespread in the northwestern and southeastern plateau and densely distributed in the Western Pamir and Nyainqêntanglha,
while they are scarce in the inner plateau; (3) the majority of rock glaciers are situated at elevations from 4,000 to 5,500 m.a.s.l.
and on slopes between 10° and 25° with north and west preferences; (4) rock glaciers show a north-west preference in the
eastern plateau, a north-only orientation in the central plateau, and no specific preference in the western plateau; (5) rock
glaciers on the Tibetan Plateau cover a total area of 6,000 km$^2$ with a mean area of 0.14 km$^2$, with rock glaciers in the western
plateau exhibiting larger areas compared to those in other areas. However, limitations inherent in imagery, deep learning
model, and manual improvement introduce uncertainties in the current inventory, which will be maintained and updated in the
future. We expect that the benchmark dataset produced by this study will facilitate the investigation into many scientific
questions related to rock glaciers and mountain permafrost on the Tibetan Plateau.

## Author contributions

YH, LL, and SH designed the study with funding from the Hong Kong Research Grants Council and the CUHK–Exeter
Joint Centre for Environmental Sustainability and Resilience. ZS and YH performed the training and prediction of deep



learning model. ZS, YH, AR, JC, XG, YH, and HY compiled the inventory. LL and XW reviewed and validated the inventory. ZS wrote the manuscript with input from YH, AR, LL, and SH.

**Competing interests**

The authors declare that they have no competing interests with respect to this work.

**Disclaimer**

Publisher's note: Copernicus Publications remains neutral with regard to jurisdictional claims made in the text, published maps, institutional affiliations, or any other geographical representation in this paper. While Copernicus Publications makes every effort to include appropriate place names, the final responsibility lies with the authors.

**Acknowledgements**

We would like to thank the Office National des Forets, Departement Service Restauration des Terrains en Montagne for providing the rock glacier inventory in the French Alps, Lingcao Huang for providing the deep learning codes, and Ho Ming Tsang for valuable assistance in the calculation of aspect angles.

**Financial support**

This work has been supported by the Hong Kong Research Grants Council, University Grants Committee (grant no. CUHK14302421, HKPFS PF20-42392), the CUHK–Exeter Joint Centre for Environmental Sustainability and Resilience (ENSURE, grant no. 4930821), the CUHK Direct Grant for Research (grant no. 4053510 and 4053592), the National Natural Science Foundations of China (grant no. 42371458, 42071410) and the "PHC PROCORE" programme (project number: F-CUHK404/22), funded by the French Ministry for Europe and Foreign Affairs, the French Ministry for Higher Education and

Research and Hong Kong Research Grants Council. AR's work is supported by the French National Research Institute for Sustainable Development (IRD), France.

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
