# Peer review of "TPRoGI: a comprehensive rock glacier inventory for the Tibetan Plateau using deep learning"

_Earth System Science Data, 2024_

## Author Comment (AC1)

The reviewer comments are in black, and our answers are in red.

This paper compiled a comprehensive inventory of rock glaciers across the entire Tibetan plateau except Himalaya and Hindu Kush. The inventory of rock glaciers is of significant importance for the study of periglacial landforms.

We thank you for commending our comprehensive inventory for its significance for periglacial studies.

However, the claim of the article is to compile an inventory of rock glaciers across the entire Qinghai-Tibet Plateau, yet it lacks a systematic inventory of the Himalayas and the Hindu Kush, where there are numerous rock glaciers and the most extensive development of such landforms. The article mentions data limitations as the reason for not completing inventories in these two regions. However, it is feasible to achieve inventory in these regions through visual interpretation using integrated data from ESRI, Bing, Google Earth, etc. Moreover, some researchers have already achieved inventory in the Himalayas (Jones). Therefore, the main issue with this inventory is its lack of completeness. If the article is to be published, the first step should be to complete the inventories in these two regions.

In section 2 Study area, lines 105-107, we specify our study area: "We selected all the 13 subregions as study areas for this work, thus covering most of the Tibetan Plateau (Fig. 1), as well as the Qaidam basin, which was not a subregion in Bolch et al. (2019b)'s study." The Himalayas and the Hindu Kush were not included in this study and this version of inventory. We have changed the phrasing from 'entire plateau' to the 'the most extensive plateau-wide inventory' consistently throughout this manuscript.

However, the inclusion of the Himalayas, the Hindu Kush, and the Tien Shan regions will be our next goal to compile a rock glacier inventory for High Mountain Asia, as is stated in section 6.3 Significance of the inventory and future work, lines 506-507: "We will extend our inventory in the future by including the Himalayas, the Hindu Kush, and the Tien Shan regions to compile a comprehensive inventory for High Mountain Asia (Figure S6)."

Other technical issues are as follows:

To include contextual information around rock glaciers, a buffer zone of 1500 meters was set. How the size of this buffer zone is chosen and its impact on model performance?

The inclusion of a buffer zone plays a pivotal role in enhancing the performance of our model by enabling it to learn contextual information surrounding rock glaciers. This augmentation facilitates more accurate boundary delineation and reduces false positives by leveraging the background samples

present in the surrounding environment (denoted as zero values in the label images). For a more comprehensive understanding of this methodology, reviewers and readers are referred to Huang et al. (2018, 2020).

The selection of an appropriate buffer zone size is equally critical. A buffer zone that is too small may fail to provide sufficient contextual information, resulting in an increased occurrence of false positives. Conversely, an overly large buffer zone risks inundating the model with excessive background information, potentially compromising its ability to accurately detect rock glaciers. In previous studies, a 300-meter buffer zone was deemed suitable for thaw slumps (Huang et al., 2020). Given the larger sizes of rock glaciers, we conducted experiments testing different buffer zone sizes ranging from 500 meters to 2000 meters. Our findings indicated that a buffer zone of 1500 meters yielded the optimal results.

We have supplemented more information in main text, section 4.1.1 Deep learning mapping, lines 194-199: "To incorporate context information from the surrounding area of a rock glacier, we established a buffer area to extract a subset of Planet images. A too small buffer area may fail to provide sufficient contextual information, resulting in an increased occurrence of false positives; while an overly large buffer area risks inundating the model with excessive background information, potentially compromising its ability to accurately detect rock glaciers (Huang et al., 2018, 2020). We conducted experiments with buffer area sizes ranging from 500 meters to 2000 meters. The results showed no significant increase in the IoU metric once the buffer area exceeded 1500 meters. Therefore, we selected 1500 meters as the buffer area."

Huang, L., Liu, L., Jiang, L., and Zhang, T.: Automatic mapping of thermokarst landforms from remote sensing images using deep learning: A case study in the Northeastern Tibetan Plateau, Remote Sensing, 10, 2067, https://doi.org/10.3390/rs10122067, 2018.

Huang, L., Luo, J., Lin, Z., Niu, F., and Liu, L.: Using deep learning to map retrogressive thaw slumps in the Beiluhe region (Tibetan Plateau) from CubeSat images, Remote Sensing of Environment, 237, 111 534, https://doi.org/10.1016/j.rse.2019.111534, 2020.

Rock glaciers are minimally represented in imagery, which theoretically poses a severe issue of data imbalance and could result in numerous false positives. This concern was also indicated by validation results. However, upon examining the prediction outcomes, there weren't too many false positives among the 48,767 candidate polygons and 44,273 rock glaciers. How was this issue addressed?

Firstly, to address the false positive issues, we supplemented our training dataset with negative samples, incorporating non-rock glacier polygons to address potential misclassifications by the deep learning model, particularly when encountering landforms exhibiting similar characteristics to rock glaciers. These non-rock glacier polygons encompass diverse features such as debris-covered glaciers, rock avalanches, and water bodies. This operation can help significantly reduce the false positives. We have added the relevant information in section 4.1.1 Deep learning mapping, lines 191-194: "We supplemented our training dataset with negative samples, incorporating non-rock glacier polygons to address potential misclassifications by the deep learning model, particularly when encountering landforms exhibiting similar characteristics to rock glaciers. These non-rock glacier polygons encompass diverse features such as debris-covered glaciers, rock avalanches, and water bodies."

Secondly, during the manual improvement phase, we not only addressed false positives through removal (the 'Remove' operation), but also rectified false negatives by adding missing detections (the 'Retrieve' operation). The results are summarized in Table 3, where the final numbers demonstrate equilibrium between the two. Hence, the number of rock glaciers predicted by the deep learning model does not exceed by far the number of eventually inventoried rock glaciers.

There are significant differences in F1 scores among different sub-regions, and although some regions with fewer rock glaciers also have large areas, how does this result support the earlier conclusion about the model having good generalization ability?

We acknowledge that the predictive performance of the deep learning model varies across different subregions, particularly in areas with large extents but few rock glaciers, such as the Tibetan Interior Mountains. These regions tend to exhibit a higher ratio of false positives over true positives, thereby potentially compromising overall accuracy. However, leveraging data from six local inventories, our deep learning model demonstrates good performance across the extensive Tibetan Plateau. Notably, in certain subregions such as Hengduan Shan, Karakoram, Nyainqêntanglha, and Western Pamir, our model achieves F1 scores of up to 0.6. This level of good performance underscores the model's commendable generalization ability, considering the inherent challenges involved in mapping rock glaciers.

In lines 75-77, the authors summarized the inventory of rock glaciers in the Qinghai-Tibet Plateau region, which contained a large amount of available inventory data (>20,000 records). However, the authors ultimately selected only a small portion of these as training data (<2,000 records). Why was this the case?

Not all rock glacier inventories are publicly available; detailed information can be found in Table S1. While some datasets include links in the 'Source of inventory dataset' column, certain extensive inventories, such as those by Zhang et al. (2021, 2023), comprising thousands of rock glaciers, remain unpublished. Consequently, our access to rock glacier datasets for training the deep learning model is restricted.

Jones et al.'s rock glacier dataset was not used, because of the incomplete boundaries (only 10% of the total 25,000 identified rock glaciers, i.e ~2,070 rock glaciers were manually digitized. This incompleteness suggests the high possibility of adjacent rock glaciers not being included in their inventory. Therefore, employing this dataset could introduce inaccuracies, as these unaccounted-for rock glaciers might be incorrectly treated as negative samples, compromising the performance of the deep learning model.

Additionally, does the training data (4,085 records) mostly consist of rock glaciers from the Alps region? Is there evidence to suggest that rock glaciers in the Alps region share similarities with those in the Qinghai-Tibet Plateau?

Referring to Table 1, approximately half of the rock glaciers included as training data analyzed are in the European Alps, with the remaining half situated in High Mountain Asia, including one inventory in the Tien Shan region. According to the IPA RGIK document, which was developed through international communication and cooperation, rock glaciers typically exhibit similar features such as front, lateral margins, and ridge-and-furrow topography. This uniformity in characteristics serves as a basis for assessing the prediction outputs from our deep learning model across the Tibetan Plateau. Given these similarities, our model demonstrates promising predictive capabilities following training with data from the Alps, and the good prediction results we have achieved can be the evidence.

In lines 107-109, please explain the unique characteristics of the Hindu Kush-Himalayas compared to other neighboring mountain ranges, as well as the differences between rock glaciers in this region and those in other areas.

The discussion on the difference in characteristics of the Hindu Kush-Himalayas compared to other neighbouring mountain ranges is beyond the scope of this study. We have deleted the relevant sentences.

In line 370, please verify the data on rock glaciers within the Qinghai-Tibet Plateau. As far as I know, multiple field expeditions have failed to find rock glaciers near the source of the Yangtze River.

Our inventory compilation is grounded in remote sensing imagery analysis, leveraging geomorphological features such as front, lateral margins, and ridge-and-furrow topography, as outlined in the International Permafrost Association (IPA) document (RGIK, 2023). Verifying each of these occurrences through fieldwork is impractical given their large number and remote locations. Therefore, our approach serves to infer the presence of rock glaciers based on remote sensing methods.

RGIK: Guidelines for inventorying rock glaciers: baseline and practical concepts (version 1.0), IPA Action Group Rock glacier inventories and kinematics, 25 pp, https://doi.org/10.51363/unifr.srr.2023.002, 2023.

In line 80, the authors emphasized the inconvenience of compiling a rock glacier inventory through visual interpretation, as it requires strong geomorphological expertise and is labor-intensive and time-consuming. However, in the end, all data were still inspected and modified through visual interpretation, making it inevitable.

The nature of labor involved in visual interpretation from scratch and our manual refinement process based on deep learning output is fundamentally distinct. While we recognize that manual validation remains necessary, it's important to highlight that the workload associated with verifying each rock glacier identified by the deep learning model is significantly less than visually examining Google Earth images across the extensive Tibetan Plateau and mapping them entirely manually.

Typically, we consider active rock glaciers as evidence of permafrost existence. However, in the southeastern part of the Qinghai-Tibet Plateau, a large number of rock glaciers seem to be outside the permafrost zone. Please explain the reliability of identifying rock glaciers in these areas.

We adopt the inventory compilation strategy recommended by the International Permafrost Association (IPA) RGIK group (RGIK, 2023), which has formalized a standard document outlining the inventorying of rock glaciers based on geomorphological evidence observed in remote sensing images, including front, lateral margins, and ridge-and-furrow topography. Additionally, given the potential inaccuracies in permafrost data, there is growing interest in utilizing rock glaciers to augment permafrost mapping efforts, as highlighted in recent studies (such as the one mentioned in the last comment: Hu et al., 2024).

RGIK: Guidelines for inventorying rock glaciers: baseline and practical concepts (version 1.0), IPA Action Group Rock glacier inventories and kinematics, 25 pp, https://doi.org/10.51363/unifr.srr.2023.002, 2023.

In Figure S4, why did this study not separate multiple rock glaciers based on other datasets?

For the first version of our inventory, our primary goal is consistency, and as such, we predominantly utilized Planet Basemap images for delineating boundaries. We intentionally avoid incorporating other datasets to prevent the introduction of inconsistencies.

The latest research progress in the Qilian Mountains region is missing. Please cite: Hu, Z., Yan, D., Feng, M., Xu, J., Liang, S., & Sheng, Y. (2024). Enhancing mountainous permafrost mapping by leveraging a rock glacier inventory in northeastern Tibetan Plateau. International Journal of Digital Earth, 17(1), 2304077.

Thanks for the information. We have added this study in our manuscript. We also update the literature review Table S1 in supplementary material and Table 5 in main text.

---

## Author Comment (AC2)

The reviewer comments are in black, and our answers are in red.

This article describes a comprehensive inventory of rock glaciers on the Tibetan Plateau using deep learning techniques. The authors also used the recent baseline and guidelines developed by the IPA Action Group Rock Glacier Inventories and Kinematics (RGIK). However, they primarily employed the geomorphological approach outlined in these guidelines, which requires strong expertise from the mappers.

The authors utilized a large volume of optical images, mainly from Planet images with a resolution of 4.7 m. The deep learning algorithm processed three bands of these Planet images, which surprisingly produced acceptable rock glacier outlines without considering other components (e.g., slope, aspect, solar radiation, surface roughness) in the model or the movement of areas (displacement). The kinematic approach, also proposed by RGIK, was not considered in delineating the rock glacier areas.

To validate the deep learning outputs, existing rock glacier inventories were used, which are assumed to be of high quality. The manuscript presents very interesting findings; however, this study needs some clarification. Therefore, I recommend that this paper undergo major revisions.

Thank you very much for your constructive comments. We have revised our manuscript accordingly and provide our point-to-point response below.

Specific comments

It seems that the deep learning algorithm is not the most efficient since it cannot support multiple datasets. What is the reason for choosing this method? What is the advantage of using it?

Deeplabv3+, the architecture adopted in this work, is a powerful and widely used deep learning model for semantic segmentation task with the capabilities of capturing multi-scale contextual information and sharp object boundaries (Chen et al., 2018). In previous studies, this model has shown good performance for mapping periglacial landforms, e.g., thaw slumps (Huang et al., 2020) and rock glaciers (Hu et al., 2023), in regional scale. Relevant information can be found in lines 178-180: "DeepLabv3+, introduced by Chen et al. (2018), was selected as the neural network architecture for the deep learning model, with Xception71 serving as its backbone (Chollet, 2017). DeepLabv3+ is specifically designed for semantic segmentation tasks and has been proven to excel in mapping permafrost landforms (Huang et al., 2020; Hu et al., 2023)."

Chen, L.-C., Zhu, Y., Papandreou, G., Schroff, F., and Adam, H.: Encoder-decoder with atrous separable convolution for semantic image segmentation, in: Proceedings of the European conference on computer vision (ECCV), pp. 801–818, https://doi.org/10.48550/arXiv.1802.02611, 2018.

Huang, L., Luo, J., Lin, Z., Niu, F., and Liu, L.: Using deep learning to map retrogressive thaw slumps in the Beiluhe region (Tibetan Plateau) from CubeSat images, Remote Sensing of Environment, 237, 111 534, https://doi.org/10.1016/j.rse.2019.111534, 2020.

Hu, Y., Liu, L., Huang, L., Zhao, L., Wu, T., Wang, X., and Cai, J.: Mapping and Characterizing Rock Glaciers in the Arid Western Kunlun Mountains Supported by InSAR and Deep Learning, Journal of Geophysical Research: Earth Surface, 128, e2023JF007206, https://doi.org/10.1029/2023JF007206, 2023.

The authors claim to have used the RGIK baselines. They mainly utilized the geomorphological approach, but it is unclear how this approach was applied to the entire inventory of 44,273 rock glaciers. Please provide more information in the text.

In section 4.1.2 Manual improvement and independent validation, lines 213-244, we describe our approach of manually compiling this inventory based on deep learning output following the RGIK baselines.

It would be beneficial to incorporate the kinematic approach (e.g., InSAR) in the near future, as both approaches are complementary.

Thanks for your suggestion. In our future work, we will use InSAR data to first validate and refine our inventory and second attribute the kinematic information following the RGIK guidelines (RGIK, 2022). We have refined the main text in section 6.3 Significance of the inventory and future work, lines 502-505: "This benchmark dataset will be maintained and updated in the future. We will leverage multi-source datasets, including InSAR data, elevation change maps from high-resolution DEM differencing, high-resolution optical images from Google Earth, ERSI basemaps, and Bing maps to validate and refine our inventory. The InSAR data will be used to attribute kinematic information (RGIK, 2022)."

RGIK: Optional kinematic attribute in standardized rock glacier inventories (version 3.0.1). IPA Action Group Rock glacier inventories and kinematics, IPA Action Group Rock glacier inventories and kinematics, p. 8 pp, 2022.

Reinosch et al. (2021) utilized InSAR time series to generate a rock glacier inventory for the western Nyainqêntanglha Range. It would be great if you could conduct a thorough comparison with this study

and others that used InSAR data (kinematic approach). This could help to understand if your results are comparable with those using other data and techniques.

In section 6.2 Comparison with existing local inventories, lines 441-476, we have compared our inventory with existing inventories on the Tibetan Plateau (including the ones developed using InSAR data (Cai et al., 2021; Reinosch et al., 2021; Hu et al., 2023; Zhang et al., 2023)) and discussed the reasons for the discrepancies between different inventories. We found that "The number of inventoried rock glaciers in our study is generally comparable to those in Daxue Shan and Hunza Basin. However, our inventory has more rock glaciers than the inventories in Gangdise Mountains and West Kunlun Shan, and fewer rock glaciers than the inventories in Nyainqêntanglha and Qilian Mountains." (lines 445-457). These discrepancies can be explained by varying operator judgement and different image sources for inventory compilation (lines 451-476).

The authors used data from only one year (2021). To me, one year is not enough to characterize such a large region. Moreover, some mass movements can be erroneously mapped, especially in areas with poor previous rock glacier inventories, making comparisons difficult. Perhaps a comparison with RGI V6 (despite its limitations) could shed some light. While including another year of data may require substantial effort, it could help remove potential discrepancies

We use Planet Basemaps images for mapping. This product is a well-processed mosaics with visual consistency and cloud mitigation by merging Planet images acquired at multiple dates. See lines 117-118: "The three-band (red, green, blue) imagery contains well-processed, scientifically accurate, and analyses-ready mosaics with a 4.77 m spatial resolution, visual consistency, and cloud mitigation (Nass et al., 2019).".

Initially, we mainly rely on Planet Basemaps mosaics from the third quarter (July-September 2021) for mapping. However, we found the product at this quarter has low quality for southeastern Tibetan Plateau (strong cloud and shadow issues). Therefore, for this region, we also used the product from the fourth quarter (October-December 2021) to improve the mapping results. Relevant information can be found in lines 120-123: "To train the deep learning model and infer new rock glaciers, we mostly utilized the Planet Basemaps mosaics from the third quarter (July-September 2021) supplemented with images from the fourth quarter (October-December 2021) when needed to mitigate image quality problems in the third-quarter images, such as shadows and image distortion."

We agree that some non-rock glacier landforms may be erroneously mapped in this version of inventory. Therefore, in our future work, we will leverage multi-source dataset, including InSAR data, elevation change maps from high-resolution DEM differencing, high-resolution optical images from

Google Earth, ERSI basemaps, Bing maps, etc. to refine our inventory. See lines 502-505: "This benchmark dataset will be maintained and updated in the future. We will leverage multi-source datasets, including InSAR data, elevation change maps from high-resolution DEM differencing, high-resolution optical images from Google Earth, ERSI basemaps, and Bing maps to validate and refine our inventory. The InSAR data will be used to attribute kinematic information (RGIK, 2022)."

Overall, no details are mentioned about the uncertainty analysis, what is the uncertainty or error estimation of the rock glacier inventory? Is it +/- 10% or >20% of the total area? Is it possible to quantify this using your methodology?

In our study, we included an independent validation (section 5.1.2 Independent validation of the inventoried rock glaciers). Based on the validation results from the two independent reviewers, approximately 87% of the rock glaciers were assigned as correct identification. Hence, the rock glacier number may have an uncertainty of around 13%.

It would be beneficial to also include other areas (Himalaya and Hindu Kush) in their study.

The inventory work for Himalaya, Hindu Kush, and Tien Shan are our future effort to create a comprehensive rock glacier inventory for High Mountain Asia. See lines 506-507: "We will extend our inventory in the future by including the Himalayas, the Hindu Kush, and the Tien Shan regions to compile a comprehensive inventory for High Mountain Asia (Figure S6)."

How did you manage the cloud cover? You mentioned that in some regions, such as Nyainqêntanglha, you found that problem

Even though the Planet Basemaps products have largely mitigated the cloud issue by merging the images from multiple dates, we still found strong cloud issues for the southeastern Tibetan Plateau in the quarter three images. Therefore, in our manual inventory phase, we also included the Planet Basemaps images from quarter four to improve the mapping results (lines 120-123: "To train the deep learning model and infer new rock glaciers, we mostly utilized the Planet Basemaps mosaics from the third quarter (July-September 2021) supplemented with images from the fourth quarter (October-December 2021) when needed to mitigate image quality problems in the third-quarter images, such as shadows and image distortion.").

185-> For the model training 70% and 30%, any specific reason for these values?

This is a typical slitting convention used in deep learning community to separate the training and validation datasets.

186-> You chose a 1,500 m buffer. Is there any technical reason for this choice?

The inclusion of a buffer zone plays a pivotal role in enhancing the performance of our model by enabling it to learn contextual information surrounding rock glaciers. This augmentation facilitates more accurate boundary delineation and reduces false positives by leveraging the background samples present in the surrounding environment (denoted as zero values in the label images). For a more comprehensive understanding of this methodology, reviewers and readers are referred to Huang et al. (2018, 2020).

The selection of an appropriate buffer zone size is equally critical. A buffer zone that is too small may fail to provide sufficient contextual information, resulting in an increased occurrence of false positives. Conversely, an overly large buffer zone risks inundating the model with excessive background information, potentially compromising its ability to accurately detect rock glaciers. In previous studies, a 300-meter buffer zone was deemed suitable for thaw slumps (Huang et al., 2020). Given the larger sizes of rock glaciers, we conducted experiments testing different buffer zone sizes ranging from 500 meters to 2000 meters. Our findings indicated that a buffer zone of 1500 meters yielded the optimal results.

We have supplemented more information in main text, section 4.1.1 Deep learning mapping, lines 194-199: "To incorporate context information from the surrounding area of a rock glacier, we established a buffer area to extract a subset of Planet images. A too small buffer area may fail to provide sufficient contextual information, resulting in an increased occurrence of false positives; while an overly large buffer area risks inundating the model with excessive background information, potentially compromising its ability to accurately detect rock glaciers (Huang et al., 2018, 2020). We conducted experiments with buffer area sizes ranging from 500 meters to 2000 meters. The results showed no significant increase in the IoU metric once the buffer area exceeded 1500 meters. Therefore, we selected 1500 meters as the buffer area."

Huang, L., Liu, L., Jiang, L., and Zhang, T.: Automatic mapping of thermokarst landforms from remote sensing images using deep learning: A case study in the Northeastern Tibetan Plateau, Remote Sensing, 10, 2067, https://doi.org/10.3390/rs10122067, 2018.

Huang, L., Luo, J., Lin, Z., Niu, F., and Liu, L.: Using deep learning to map retrogressive thaw slumps in the Beiluhe region (Tibetan Plateau) from CubeSat images, Remote Sensing of Environment, 237, 111 534, https://doi.org/10.1016/j.rse.2019.111534, 2020.

405-> "However, rock glaciers are frequently found in regions characterized by poor image quality due to factors associated with cloud cover, shadows, and distortions, which are common in

mountainous areas". This is why it would be useful to incorporate more than one year of data into your dataset

In our study, we incorporate the Planet Basemaps images from both quarter three (July-September 2021) and quarter four (October-December 2021) for mapping. Since the Planet Basemaps images are processed mosaics by merging many Planets images from different dates for cloud mitigation and improvement of other issues, it can be regarded we use many images from different dates for mapping.

---

## Referee Report (RR1)

The proposed paper presents a data set containing a rock glacier inventory (RoGI) in the Tibetan Plateau. The vastness of the study area combined with the number of rock glaciers that have been inventoried are the main, bot not the only, arguments that make this data set to be of significant importance for the scientific community.

The methodology used to compile the inventory complies with good practices, recommendations and the latest guidelines in rock glacier inventorying as proposed by the Rock Glacier Inventories and Kinematics community (RGIK).

Although, it doesn't use a specific kinematic method (e.g. remote sensing, GNSS), the geomorphological approach in estimating the activity status of RGs is an accepted method by the scientific community.

The performance of the deep learning algorithm varies significantly between subregions, performing poorly in at least five of them, but the manual revise of the deep learning mapping solves this shortcoming.

Thus, the resulting data set is rigorous enough to be considered for publication. As for the manuscript itself, I recommend some minor revisions and clarifications.

Specific comments:

Line 238 – 240

Please explain in more detail how the "Retrieve" operation in the "Manual improvement and independent validation" was performed. Specifically in what areas were the rock glaciers added, as I assume you didn't check the entire study area.

Line 251 – 258

It is not clear if the two reviewers have made changes to the RoGI or if their only purpose was to evaluate the accuracy of the inventory. In other words, a rock glacier that was drawn by the seven mappers and the which was found to be incorrectly identified by the reviewers is still part of the inventory? Please clarify.

Paragraph 6.1.2 Limitations of the deep learning model (lines 408 – 427)

It is important to acknowledge the limits of the model as they are the based on which the model can be improve in the future. And the most important model limitation, in my opinion, is the number of bands that can be used as input data. The use of morphometric data (e.g. slope, terrain roughness) and lithological data might significantly increase the model accuracy. Please consider acknowledging this limitation or have an argument on why this is not a significant limitation.

Table 3 (page 15)

The first two data columns are expressed in the number of rock glaciers, while the following three are expressed as the total areas of rock glacier. If there are rock glaciers that greatly vary in size, then some

metrics (e.g. F1 score) might be influenced by it. Also, tables 4 and 5 are expressed in number of rock glaciers. Please explain why you choose to have this inconsistency or consider revising the figure in order that all the data columns are expressed in the same unit. Consistency is important for accurate assessment and for easy reading.

---

## Author Response (AR2)

The reviewer comments are in black, and our answers are in red.

The proposed paper presents a data set containing a rock glacier inventory (RoGI) in the Tibetan Plateau. The vastness of the study area combined with the number of rock glaciers that have been inventoried are the main, bot not the only, arguments that make this data set to be of significant importance for the scientific community.

We thank the reviewer for recognizing the scientific significance of our inventory.

The methodology used to compile the inventory complies with good practices, recommendations and the latest guidelines in rock glacier inventorying as proposed by the Rock Glacier Inventories and Kinematics community (RGIK).

Although, it doesn't use a specific kinematic method (e.g. remote sensing, GNSS), the geomorphological approach in estimating the activity status of RGs is an accepted method by the scientific community.

Estimating the activity status of rock glaciers is our future effort. We will use InSAR data to attribute the kinematic information. See lines 512-513: "The InSAR data will be used to attribute kinematic information (RGIK, 2022)."

The performance of the deep learning algorithm varies significantly between subregions, performing poorly in at least five of them, but the manual revise of the deep learning mapping solves this shortcoming.

Yes, manual improvement is crucial for producing this inventory.

Thus, the resulting data set is rigorous enough to be considered for publication. As for the manuscript itself, I recommend some minor revisions and clarifications.

Specific comments:

Line 238 – 240

Please explain in more detail how the "Retrieve" operation in the "Manual improvement and independent validation" was performed. Specifically in what areas were the rock glaciers added, as I assume you didn't check the entire study area.

The reviewer is right. We didn't check the entire study area but focused on areas in proximity to the polygons identified by the deep learning model. Specifically, when we observed a missing rock glacier within the window scope during the manual inspection of candidate rock glacier polygons, we added it to our inventory.

Relevant information has been added to the manuscript in lines 240-241: "When missing rock glaciers were identified during the manual inspection of nearby candidate polygons, they were added to the inventory.". and lines 446-448: "Additionally, the "Retrieve" operation focused on areas where missing rock glaciers were observed near the polygons identified by the deep learning model. Consequently, some rock glaciers may have been missed without conducting an exhaustive examination of the entire study region.".

Line 251 – 258

It is not clear if the two reviewers have made changes to the RoGI or if their only purpose was to evaluate the accuracy of the inventory. In other words, a rock glacier that was drawn by the seven mappers and the which was found to be incorrectly identified by the reviewers is still part of the inventory? Please clarify.

The two reviewers have not made changes to the RoGI. Their only purpose was to evaluate the accuracy of the inventory. We have included an attribute "ADDI_INF" in the inventory dataset to provide this information. See lines 267-268: "The ADDI_INF provides information on whether the rock glacier has been recognized as a false identification by the reviewers.".

Paragraph 6.1.2 Limitations of the deep learning model (lines 408 – 427)

It is important to acknowledge the limits of the model as they are the based on which the model can be improve in the future. And the most important model limitation, in my opinion, is the number of bands that can be used as input data. The use of morphometric data (e.g. slope, terrain roughness) and lithological data might significantly increase the model accuracy. Please consider acknowledging this limitation or have an argument on why this is not a significant limitation.

Thanks for pointing it out. Yes, it is an important limitation of our model and hasn't been discussed in the current manuscript. We have added this in our revised manuscript. See lines 417-420: "A key limitation of the current deep learning model is the restricted number of input bands. Our model only utilizes RGB bands, while inherently excluding crucial topographic information such as slope and elevation. As rock glacier occurrence is closely related to topography and underlying geology, the

absence of morphometric inputs like terrain roughness and slope, as well as lithological data, may hinder the model performance.".

Table 3 (page 15) The first two data columns are expressed in the number of rock glaciers, while the following three are expressed as the total areas of rock glacier. If there are rock glaciers that greatly vary in size, then some metrics (e.g. F1 score) might be influenced by it. Also, tables 4 and 5 are expressed in number of rock glaciers. Please explain why you choose to have this inconsistency or consider revising the figure in order that all the data columns are expressed in the same unit. Consistency is important for accurate assessment and for easy reading.

Thanks for the comment. The three tables serve different purposes, thus may have some inconsistencies. Table 3 aims to evaluate the performance of our deep learning model. Here, we chose to evaluate the area instead of counting numbers, because the area-based evaluation is more relevant to the pixel-based evaluation of a deep learning model.

However, for Tables 4 and 5, the evaluation and comparison are conducted on the inventory rather than deep learning model. For evaluating an inventory, as we pointed out in lines 252-253: "Given the difficulty in accurately evaluating the delineated boundaries, our validation focused primarily on verifying the primary markers.". Thus, it is more practical to compare the numbers instead of the areas for these two tables.